# Hypoxic Radioresistance: Can ROS Be the Key to Overcome It?

**DOI:** 10.3390/cancers11010112

**Published:** 2019-01-18

**Authors:** Hui Wang, Heng Jiang, Melissa Van De Gucht, Mark De Ridder

**Affiliations:** Department of Radiotherapy, Universitair Ziekenhuis Brussel, Vrije Universiteit Brussel, Laarbeeklaan 101, 1090 Brussels, Belgium; hui.wang.ecnu@gmail.com (H.W.); jiangheng1981@gmail.com (H.J.); Melissa.Van.De.Gucht@vub.be (M.V.D.G.)

**Keywords:** reactive oxygen species, radiation, hypoxia, radiosensitization

## Abstract

Radiotherapy is a mainstay treatment for many types of cancer and kills cancer cells via generation of reactive oxygen species (ROS). Incorporating radiation with pharmacological ROS inducers, therefore, has been widely investigated as an approach to enhance aerobic radiosensitization. However, this strategy was overlooked in hypoxic counterpart, one of the most important causes of radiotherapy failure, due to the notion that hypoxic cells are immune to ROS insults because of the shortage of ROS substrate oxygen. Paradoxically, evidence reveals that ROS are produced more in hypoxic than normoxic cells and serve as signaling molecules that render cells adaptive to hypoxia. As a result, hypoxic tumor cells heavily rely on antioxidant systems to sustain the ROS homeostasis. Thereby, they become sensitive to insults that impair the ROS detoxification network, which has been verified in diverse models with or without radiation. Of note, hypoxic radioresistance has been overviewed in different contexts. To the best of our knowledge, this review is the first to systemically summarize the interplay among radiation, hypoxia, and ROS, and to discuss whether perturbation of ROS homeostasis could provide a new avenue to tackle hypoxic radioresistance.

## 1. Introduction

Radiotherapy provides local control and cure for many tumor types by using high-energy rays. Since the discovery of X-rays in 1895, X-rays, γ-rays and electron beams, the so-called low linear energy transfer (LET) radiation, have been widely used in the management of malignant tumors [1]. Later, other types of rays, such as proton and carbon ions, which have high LET radiation and are able to more effectively kill hypoxic cells, have been applied gradually in clinics [2,3,4]. However, owing to the high construction and operation costs of the accelerator system, only a limited number of patients can get access to high-LET radiation, and low-LET radiation remains the most prevalent in radiotherapy. In this review, radiotherapy is, therefore, referred to low-LET radiation.

Hypoxia is a common feature of solid tumors, resulting from the imbalance between oxygen availability and consumption, and it is defined as one of the most important causes for radiotherapy failure since 1953 [5,6]. After six decades of intensive research, numerous strategies have been developed to overcome hypoxia-induced radioresistance, including improving tumor oxygenation by hyperbaric oxygen, oxygen mimetic radiosensitizers, and hypoxia-selective cytotoxins [7]. Nevertheless, their adequate clinical use is compromised so far by both limited efficacy and side effects.

Reactive oxygen species (ROS), usually upregulated in the tumor cells and tumor environment, are the effector molecules of radiation, contributing to radiation-induced DNA damage and cancer cell death [8,9,10]. Enhancing ROS production by various means has been widely investigated as a radiosensitizing strategy with promising results primarily generated in aerobic conditions [11,12]. The fundamental feature of hypoxia is a shortage of oxygen, so there was a common notion that hypoxic cells are characterized with less ROS (oxygen is the substrate of ROS) and resistant to pharmacological ROS insults. Due to this, only a few ROS insults are studied as hypoxic radiosensitizers [13,14,15,16,17,18,19,20]. Paradoxically, evidence from a large number of studies using diverse methods demonstrates that more ROS are released in hypoxic than in normoxic cells [21,22,23,24,25]. This raises the question regarding how cancer cells survive under hypoxia-induced oxidative stress. It has been demonstrated that the antioxidant defense network is more mobilized in hypoxic state and renders cancer cells rapidly detoxify ROS [26,27]. As such, hypoxic cells rely heavily on the antioxidant system to effectively keep ROS below lethal values, thus making them also vulnerable to any ROS insults that impair links involved in sustaining ROS balance. Given hypoxic tumor cells show elevated activity of antioxidant systems, measures to disrupt ROS homeostasis could be an attractive approach to selectively enhance hypoxic radioresponse. In this review we, therefore, discuss the biological action of radiation, the interaction among radiation, hypoxia, and ROS, and end it with evidence that overcoming hypoxic radioresistance could be achieved by disruption of ROS homeostasis.

## 2. Biological Action of Radiation

The biological changes caused by radiation to the targeted tissues are initiated by an absorption process, in which the energy of radiation is deposited into the encountered molecules on its path [28]. Briefly, radiation interacts with atoms of the absorbed tissue and gives its energy away. As a consequence, according to Compton process, an electron is ejected from the atom and further reacts with other molecules. This chain of events results in the break of vital chemical bonds that culminates in biological changes. 

DNA is a pivotal molecule that stores biological information and decides the fate of the cells, and it is the primary target of radiation and its damages are the prime source of biological effects of radiation [8,9,10]. Electron resulting from absorption of radiation directly interacts with DNA, causing DNA lesions such as single- and double-strand breaks. It can also interact with other molecules to generate free radicals that, in turn, cause DNA damage, which plays a leading role in radiation-induced biological effects [28]. The effect of radiation on DNA is illustrated in Figure 1. A free radical is an atom or molecule carrying an unpaired orbital electron in the outer shell. ROS are oxygen-containing unstable chemical species that easily react with other molecules [29]. The majority of ROS are free radicals, such as superoxide anions (O_2_^−^) and hydroxyl radicals (HO). While some of ROS are non-radical species, and typical examples of non-radical species are hydrogen peroxide (H_2_O_2_) and nitric oxide (NO) [12]. Of note, the greater part of free radicals generated by radiation are ROS [28]. 

About 60% of a human body and about 80% of a cell is composed of water; conceivably, radiation predominantly interacts with water molecules to generate free radicals in the targeted tissue [12,30]. This reaction consists of two steps: water molecules first become ionized by radiation to generate H_2_O^+^ and a free electron; then, H_2_O^+^ as an ion and free radical has an extremely short lifetime, reacting with another water molecule soon after its generation to form HO^·^. Although HO^·^ is highly reactive, it can diffuse a short distance to reach critical cellular molecules to induce biological effects or convert into other free radicals or ROS for further reactions. ROS generated by radiation is the essential contributor to DNA damage, ascribing to about two-thirds of DNA damage in mammalian cells, and this effect can be severely impaired by ROS scavenger [28]. 

The physical and chemical reactions initiated by radiation takes less than a millisecond. However, the expression of biological effect may be hours, days, and months because cell death-induced by radiation often occurs when the damaged cell attempts to divide [28]. For example, radiation-induced damage to the epidermal layer of the skin and gastrointestinal epithelium appears a few days or weeks, while the damage to slowly proliferating tissues, such as the central nervous system and heart, emerges after a delay of months or years. In addition, the extent of response to radiation varies in each individual due to both cellular and microenvironmental factors, including the intrinsic capacity of cells to repair the DNA damage and to detoxify ROS, the ability of immune surveillance to attack the residual tumor cells, and the degree of oxygen levels the cells located. All these factors differ in individuals, so in patients with the same type and similar stage of cancer, they respond differently to radiation with diverse local control and outcome.

## 3. Hypoxia

### 3.1. Tumor Vasculature and Hypoxia

In the tumor, the neovasculature is often sparse and morphologically and functionally abnormal, leading to severe deficiencies in the perfusion of oxygen and nutrients [13]. For example, blood vessels in tumors often present “blind-ends” and temporary occlusions. Altogether, these abnormalities lead to the heterogeneous hypoxic microenvironment, a hallmark of solid tumors [31,32].

It is important to recognize that hypoxia in tumors can result from two quite different mechanisms: diffusion-limited or chronic hypoxia [33] and perfusion-limited or acute hypoxia [34]. These mechanisms are not mutually exclusive but, on the contrary, typically operate together to cause low and fluctuating oxygen levels. Diffusion-limited or chronic hypoxia is caused by metabolic oxygen consumption that exceeds oxygen supply through the vasculature. Therefore, cells lying near the capillaries within the diffusion distance of oxygen (<100 µm) are well oxygenated; cells lying at the edge of the diffusion distance are chronically hypoxic yet viable; and cells located distant from the capillaries, beyond the diffusion distance of oxygen become necrotic [13]. Cells that become hypoxic in this way remain hypoxic for a long period of time (from a few hours to many days) until they die and become necrotic. Perfusion- limited or acute hypoxia is caused by the temporary closing of a tumor blood vessel owing to the malformed vasculature of the tumor, which lacks smooth muscle and a basement membrane. The cells are intermittently hypoxic because normoxia is restored each time when the blood vessel opens up again. Both chronic diffusion-limited and acute perfusion-limited hypoxia can cause a topographically defined cellular subpopulation to be protected at the time of radiation without being killed by severe oxygen starvation.

### 3.2. Mechanisms of Hypoxic Radioresistance

The response of cells to ionizing radiation is strongly dependent upon the presence of oxygen, and hypoxia leads up to 3 times radioresistance [28]. Different mechanisms have been suggested to interpret this phenomenon; among them, the oxygen fixation hypothesis is the best accepted [6], which is illustrated in Figure 2. Radiation induces ionization in or close to the genomic DNA of target cells and produces various radicals, which cause DNA strand breaks. Oxygen, being the most electron-affinic molecule in the cell, reacts extremely rapidly with the free radicals and makes the damage permanent. In absence of oxygen, the DNA radicals are reduced by compounds containing sulfhydryl groups (SH groups), which repair the DNA to its original form. In a sense, oxygen may be said to “fix” or make the radiation lesion permanent, known as the oxygen fixation hypothesis. 

There are other mechanisms involved in hypoxic radioresistance. Hypoxia-inducible factor (HIF-1) is the best-characterized transcription factor mediating hypoxic response, which consists of an inducible alpha subunit and a constitutively expressed beta subunit [35]. In normoxia, HIF-1α is hydroxylated and tagged by oxygen-dependent prolyl hydroxylases (PHD), allowing binding to von Hippel- Lindau (VHL) complex for proteasomal degradation. While in hypoxia, the activity of PHD is inhibited, allowing HIF-1α to accumulate and dimerize with HIF-1β subunit to bind to targeted genes and enhancing their transcription. HIF-1 regulates more than a hundred genes and confers radioresistance by acting upon multiple mechanisms at different levels [36]. For example, HIF-1 enhances the activity of glycolysis, serine synthesis pathway, and pentose phosphate pathways, which in turn increase the production of antioxidants and thus buffering radiation-induced ROS and causing radioresistance [37,38,39,40]. In addition, hypoxia itself elevates ROS production which, in turn, (1) triggers a feedback loop to stimulate metabolism that is in favor of generation of antioxidant [41,42] and (2) activates autophagy to accelerate the clearance of cellular ROS products, making cells radioresistant [26,43]. Moreover, hypoxia sustains a “quiescent” state of stem cells preserving their potential to proliferate and differentiate, thus protecting them from radiotherapy [44]. 

### 3.3. Tumor Hypoxia and Radiotherapy Outcome

Since the observation of the oxygen effect, a variety of techniques have been used to determine the oxygenation of human tumors. Among these techniques, a polarographic oxygen electrode is considered the “gold standard” for measuring tumor pO_2_. The data from polarographic oxygen electrode studies indicate that hypoxia can be used to predict radiotherapy outcomes for a variety of tumors, including cervix carcinoma, head-and-neck tumors, and soft tissue sarcomas. As summarized in Table 1, these investigations indicate that cervix carcinomas and head-and-neck cancers are poorly oxygenated, and the oxygenation status in all three different cancers is an independent, adverse prognostic factor for radiotherapy [45,46,47,48,49,50,51,52,53,54,55,56]. Thus, detection of hypoxia in the clinical setting may, therefore, be helpful in selecting high-risk patients for individual and/or more intensive treatment schedules. 

Although the most solid evidence of tumor hypoxia in patients is derived from a polarographic oxygen electrode, this approach is limited to accessible tumors that are suitable for electrode insertion. Hence, non-invasive approaches, particularly image-based modalities, have been investigated. The nitroimidazole family of compounds have been previously developed as hypoxic radiosensitizers. Since nitroimidazole-based drugs are able to accumulate in hypoxic cells, they have been repurposed as hypoxia tracers or probes that are detected by positron emission tomography (PET) imaging or immunohistochemistry. A great number of nitroimidazole-based PET tracers have been developed, such as 18F-fluoromisonidazole (18F-MISO), 18F-fluoroazomycin-arabinozide (18F-FAZA), and 18F-2-nitroimidazol-tri-fluoropropyl acetamide (18F-EF3) [57]. In line with the polarographic oxygen electrode, by using immunohistochemistry or PET, hypoxia is detected in sarcomas and head and neck carcinomas, as well as in lung adenocarcinomas and breast cancer, correlated with poor therapeutic responses [36,58,59,60]. In addition to PET, non-invasive magnetic resonance (MR) techniques, including MR imaging (MRI) and MR spectroscopy (MRS), have been exploited to monitor tumor hypoxia [61]. Among different subtypes of MRI and MRS techniques, such as dynamic contrast-enhanced (DCE), blood oxygen level dependent (BOLD), diffusion-weighted (DW) MRI, ^31^P-MRS, and ^1^H-MRS, DCE–MRI clearly demonstrates the correlation of hypoxic state in tumors with radiotherapy outcome in patients with cervical cancer [62,63,64]. Recently, by exploiting the intravoxel incoherent motion signal in DW-MRI, both oxygen consumption and supply can be assessed, and the generated images on hypoxic tumors could help identify aggressive disease in prostate cancer [65]. Tumor hypoxia can also be inferred from the expression of various endogenous proteins, such as HIF-1α, glucose transporters-1 or -3, vascular endothelial growth factor-A, and carbonic anhydrase-9 [66,67,68,69,70,71].

## 4. ROS

Cells produce ROS at diverse sites, including mitochondria, endoplasmic reticulum, peroxisomes, and the family of Nicotinamide Adenine Denucleotide Phosphate Reduced Form (NADPH) oxidases [29]. The largest contributor to cellular ROS is the mitochondria, accounting for about 90% of the total cellular ROS generation [12,72,73]. In mitochondria, the three best-characterized spots for ROS generation are complex I, II, and III within the mitochondrial electron transport chain [74]. 

As a metabolic byproduct and highly reactive molecules, ROS are constantly generated inside the cells and react with other molecules acting as secondary messengers to modulate biological functions. ROS involve in cell proliferation, differentiation, autophagy, and adaptation to hypoxic, metabolic, and immune stresses [12,75,76]. While excessive production of ROS causes damage to critical cellular components, for example, DNA, RNA, and proteins, resulting in cell damage or even death. Thus, in normal physical condition, the intracellular ROS production is rigidly monitored and regulated by antioxidant systems consisting of antioxidants and enzymes. When the balance between ROS production and elimination is lost, a condition known as oxidative stress occurs, leading to cytotoxicity, genotoxicity, and carcinogenesis [12,28]. 

About two-thirds of radiation-induced DNA damage is caused by ROS, thus, the capacity of cells to detoxify ROS inevitably impacts on the radiosensitivity of tumor cells [12]. Increased expression or activity of antioxidant enzymes, such as glutathione S-transferase, glutathione reductase, and peroxiredoxin, is correlated with poor radioresponse in patients [77,78,79]. In addition, to cope with radiation-induced oxidative stress, adaptive cascades reactions are triggered to further elevate the capacity to detoxify ROS. In preclinical models, radiation evokes an upregulation of expression or activity of redox enzymes in cancer cells, such as sodium dismutase and catalase [12,80,81]. In patients with oral cancer, the expression of antioxidant enzymes such as sodium dismutase is upregulated after radiation [82,83].

## 5. The Interplay between Hypoxia and ROS

### 5.1. Hypoxia Enhances ROS Production

It would seem improbable that a decrease in oxygen, a substrate for ROS, would cause an increase in ROS. However, many studies have reported that chronic hypoxia elicits an increase in oxidant production in both primary and malignant cells [21,23,24,25,84,85,86]. In primary cardiomyocytes, the upregulation of ROS is graded to the severity of hypoxia, such that greater increases were seen with 1% O_2_ compared with 3 or 5% [85]. Similarly, hypoxia enhanced the ROS production in primary pulmonary artery smooth muscle cells, and the addition of ROS scavenger or reinforcement of antioxidant enzymes expression attenuates ROS generation [86]. This phenomenon is observed in malignant cells as well, in acute myeloid leukemia (HEL, HL60-VCR), the levels of ROS are progressively augmented by the duration of chronic hypoxia up to 72 h [84]. In hepatocarcinoma cells (HepG2, SMMC-7721, and Huh7), the ROS levels rise after hypoxic stress [23]. Correspondingly, activity of glutathione system and levels of cofactor NADPH are enhanced to keep ROS below lethal values, which could be due to the downstream effect of ROS induced stabilization of HIF-1α [24]. In addition, in breast cancer cells (MDA-MB-468), hypoxia increases the intracellular levels of ROS, leading to upregulation in N-cadherin and SERPINE1, two proteins involved in cell adhesion [25]. Of note, changes that adjust to chronic hypoxia do not always lead to increased generation of ROS, it could result in a decline of ROS in some cancer cells. For example, in colorectal cancer cells (HCT-116, HT-29, and LoVo), compared with the normoxic condition, the ROS level was reduced in lysate from all of the three cell lines under hypoxia [87]. Thus, this phenomenon should be interpreted cautiously in different contexts. 

Although the specific mechanism has not been described, a likely source of ROS production during hypoxia appears to be mitochondria, more particularly complex III. The ubisemiquinone radical is repeatedly generated at both the Qo and Qi sites of complex III during the electron transport process. Molecular oxygen is highly electrophilic and can potentially capture the electron from ubisemiquinone [21]. Under hypoxic condition, the lifetime of ubisemiquinone is prolonged, creating more opportunity for oxygen to react with the electron to generate ROS [22]. In line, mitochondrial inhibitor antimycin A that acts at the Qi site of complex III prolongs the lifetime of ubisemiquinone, resulting in ROS production in hypoxia [24]. Beside mitochondria, nitric oxide synthases (NOS), and NADPH oxidase have also been implicated as contributors to increased ROS production in hypoxia [26,88,89]. NO and its derivatives are a specific group of ROS, playing important roles in different physiological and pathological conditions, such as neurotransmission and vasodilation. In mammals, NO is synthesized by a family of enzymes referred as NOS. The inducible NOS (iNOS) is a hypoxia response gene; so under chronic hypoxia, generation of NO is significantly increased in cancer cells [13] and involved in the adaptation of cells to hypoxic condition [90].

Increased ROS production during chronic hypoxia remains debatable given paradoxical findings, while acute hypoxia stimulating ROS production has been widely studied and accepted due to its involvement in myocardial injury [91]. Acute hypoxia comprises the phases of hypoxia (ischemia) and reoxygenation (reperfusion). In the phase of hypoxia, production of ROS is increased along with a decrease in cellular antioxidants that is similar to the chronic hypoxia. However, in the phase of reoxygenation, the production of ROS upon acute reintroduction of oxygen is boosted into a much higher level that is sufficient to induce cell damage and even death [91,92,93], regarded as the causal factor for ischemia-reperfusion injury. In hepatoma cells (HepG2 and Hep3B), the intracellular ROS accumulate during both hypoxia and reoxygenation and at a much faster rate during reoxygenation [94]. Furthermore, excessive production of ROS triggers autophagy to clear damaged cellular components [94]. In breast cancer cells (MDA-MB-231), a significant increase in ROS levels over normoxic cells is observed after 4 h of reoxygenation along with high levels of thioredoxin [95]. In glioblastoma cells (GBM8401 and U87) and xenografts, ROS levels are upregulated under acute hypoxia, concomitant with increased tumor cell growth in vitro and in vivo [96]. Based upon these observations, it has been suggested that ROS released during hypoxia act as signaling molecules that trigger diverse functional responses, such as autophagy and cell migration, to make cancer cells adapt or escape from the deteriorated environment before oxygen availability becomes very limiting [21].

### 5.2. ROS Mediates Hypoxia Adaptation

Hypoxia enhances ROS production; reciprocally, ROS assist tumor cells to adapt to hypoxia via stabilization of HIF-1α [22]. HIF-1 has multiple functions. With respect to ROS homeostasis, HIF-1 regulates metabolic reprogramming to improve ROS buffering capacity, or directly increase the expression of genes implicated in antioxidants production [27,97]. Thus, knockout of HIF-1 could lead to the death of hypoxic tumor cells due to overwhelming levels of intracellular ROS [98,99]. The site of ROS that contributes to stabilizing HIF-1α is shown as mitochondria complex III [100]. However, later, the same effect is observed even bypassing complex III [101], indicating that ROS stabilizing HIF-1 α is not limited to complex III alone but dependent on the mitochondrial electron transport chain as a whole [35]. How ROS induce HIF-1α stabilization or accumulation is elusive. It was speculated that it is through inhibition of PHD enzymes and thus preventing HIF-1α from degradation; however, this hypothesis is turned down by the evidence that ROS do not regulate PHD activity directly. Later, evidence points out that hypoxia-triggered ROS induce PI3K/AKT pathway and ERK phosphorylation that in turn increase HIF-1α transcription and translation, indicating that ROS-mediated increase of HIF-1α expression is the mechanism [35,102,103].

In addition, hypoxia activates iNOS and increases NO generation, which in turn results in HIF-1α accumulation [90]. Of note, the effect of endogenous NO on the HIF-1 accumulation occurs only at high concentrations (μM); at lower levels (nM), oppositely, endogenous NO inhibits HIF-1α stabilization under hypoxia [35,90]. The mechanisms of NO-mediated HIF-1α accumulation in hypoxia are not yet clarified. Possibly, it is through post-translational modification of HIF-1α protein or by inhibiting PHD activity [104,105,106], however, more studies are required to support these speculations. 

## 6. Overcoming Hypoxic Radioresistance by Disruption of ROS Homeostasis

After decades of investigation, hypoxia remains one of the greatest obstacles to improve cancer response to radiotherapy. Hyperbaric oxygenation and oxygen-mimetic nitroimidazoles have been developed and proven to significantly improve local tumor control in different types of cancer [6,7]. Nevertheless, these hypoxic modifications have not found a place in routine clinical practice due to their inconvenient applications or/and neurotoxicity [7]. Hence, studies are ongoing to explore any possible approaches to overcome hypoxic radioresistance. In this context, the growing knowledge of ROS and hypoxia may offer us new possibilities, as illustrated in Figure 3 and demonstrated in Table 2.

### 6.1. Hypoxic Radiosensitization by NO

As early as in 1957, NO gas was shown to efficiently radiosensitize hypoxic bacteria to ionizing radiations [151]. However, it was not until the early 1990s that the radiosensitizing property of NO was revisited. By using NO-releasing agents (diethylamine nonoate and S-nitrosoglutathione), the hypoxic radioresistance of Chinese hamster V79 lung fibroblast is almost completely abolished [107,108]. Subsequently, by using NO donors, such as nitroglycerin [152], spermine nonoate, and sodium nitroprusside, similar hypoxic radiosensitizing effects are observed in different types of cancer cells [110,111]. This effect is verified in early clinical trials by using NO donor nitroglycerin; in rectal, prostate, and non-small lung cancer patients, radiotherapy combined with nitroglycerin demonstrates an acceptable toxicity profile [109,153,154]. 

To achieve a more specific and localized generation of NO and prevent systemic side effects, endogenously produced NO has been studied to overcome hypoxic radioresistance. Compared with chemical NO donors, NO produced inside tumor cells or in co-cultured normal cells (macrophages or hepatocytes) enhances hypoxic radioresponse at 10 to 30-times reduced extracellular levels of NO, providing a favorable profile of NO-related cytotoxicity [112,155,156]. Next to iNOS, another isoform endothelial NOS (eNOS) is capable to modify tumor radioresponse. Insulin as an inducer of eNOS is shown to increase both tumor oxygenation and radioresponse in a liver and fibrosarcoma mouse tumors [113]. 

NO modulated radiosensitization although still need to be studied in details, several mechanisms have been unraveled to underlie the effects: (1) fixation of radiation-induced DNA damage, which is reported for NO gas, NO donors, and iNOS in vitro [111,151]; (2) vasodilating and thus improving tumor perfusion and oxygenation, which is established for some bioreductive NO donors and eNOS in vivo [157,158,159]; and (3) inhibition of tumor cell respiration and oxygen sparing, which is demonstrated for some bioreductive NO donors and eNOS, and confirmed in ex vivo (isolated) tumor cells [113,160,161].

### 6.2. Hypoxic Radiosensitization by Inhibition of Antioxidant Enzymes

Antioxidant systems are central to sustain the ROS balance in tumor cells, and their dysregulation is attributable to hypoxic radioresistance. Inhibition of antioxidant proteins therefore could be effective to counteract hypoxia-induced radioresistance [12].

Buthionine sulphoximine (BSO) is a classical drug used to inhibit glutamate-cysteine ligase that is the rate-limiting enzyme in the production of antioxidant glutathione synthesis. Glutathione levels are higher in hypoxic than in non-hypoxic regions, and the treatment with BSO produces a more pronounced glutathione depletion in regions of hypoxia [162]. BSO alone although exhibits a marginal effect to enhance radioresponse of hypoxic tumor cells [114], in combination with hypoxic radiosensitizers, such as misonidazole and SR-2508, a synergistic effect to enhance radioresponse of the hypoxic tumor is detected, correlating with DNA strand breaks and base damage [114,115]. Early clinical trials (phase I and II) demonstrate that at tolerable doses, BSO administration reduces the level of glutathione in both tumor samples and blood lymphocytes [12,163,164,165].

Dimethylfumarate (DMF) and diethylmaleate (DEM) form covalent bonds with glutathione, consequently, deplete biologically active glutathione [116]. In Chinese hamster ovary cells, DMF depletes glutathione to less than 10% of control, leading to a significant enhancement of hypoxic radiosensitivity [16]. Likewise, DEM enhances radiosensitivity of hypoxic tumor cells and enhance the radioresponse of the tumor [17]. Similarly to BSO, the action of DMF and DEM can synergize with oxygen-mimetic radiosensitizers, such as misonidazole, owing to altering the metabolism of the drugs and then potentiating their effects [116]. 

Piperlongumine (PL), naturally synthesized in long pepper, is able to perturb ROS homeostasis by inhibition of glutathione S-transferase and thioredoxin reductase [166,167]. These two enzymes play important roles in sustaining the activity of antioxidants: glutathione S-transferase catalyzes the conjugation of glutathione with its substrate; thioredoxin reductase maintains the reduced form of antioxidant thioredoxin. For hypoxia-tolerant lung cancer cells, PL treatment induces overproduction of ROS, subsequently overcoming radioresistance and delaying tumor growth [18,117].

Auranofin (AF) is a well-characterized irreversible thioredoxin reductase inhibitor [168]. In breast cancer cells, AF overcomes hypoxic radioresistance with mechanism linked to ROS-mediated mitochondrial dysfunction, DNA damage, and apoptosis [20]. This effect could be further amplified by combining with BSO, leading to significant tumor growth delay and increased the survival rate of tumor-bearing mice [19,20]. Currently, several clinical trials are initiated to evaluate the safety and therapeutic effect of AF as monotherapy or in a combined regimen, and the results are awaited (NCT03456700, NCT01737502, NCT01747798). 

### 6.3. Hypoxic Radiosensitization by Inhibition of HIF-1

Due to its critical function in promoting tumor cell adaptation to microenvironmental stress, HIF-1 has been recognized as an excellent molecular target to overcome cancer cell radioresistance [128,169]. Silencing or pharmacological inhibition of HIF-1 indeed increases sensitivity to radiation in diverse tumor models. HIF-1 knockdown in human hepatoma cells inhibits proliferation, induces apoptosis and promotes radiosensitivity in chemically-induced hypoxia [118]. In the prostate cancer cell line, the knockdown of HIF-1 by siRNA induces apoptosis and G2/M cell cycle arrest, resulting in radiosensitization [119]. In FaDu and ME180 xenograft tumors, blocking the HIF1 response during transient hypoxic stress increases hypoxia, reduces lactate levels and enhances response to high-dose single-fraction radiation [170]. In laryngeal carcinoma, simultaneous inhibition of HIF-1α and glucose transporter-1 expression increases the radiosensitivity, decreases microvessel density, and promotes apoptosis and necrosis [171].

Many compounds have been reported to enhance radioresponse via inhibition of HIF-1, for example, SN-38 (the active metabolite of irinotecan) [120], atorvastatin (a lipid-lowering agent) [121], NSC74859 (a STAT3 inhibitor) [122], and berberine (a naturally compound) [123]. So far, the most studied HIF-1 inhibitors in this context are YC-1 and PX-478 [128,129]. YC-1 was at first synthesized with the aim of activating soluble guanylate cyclase and inhibiting platelet aggregation; later, it was proved to inhibit HIF-1 via induction of HIF-1α protein degradation and inhibition of HIF-1α translation [172,173,174]. YC-1 is shown to enhance the radioresponse of lung cancer cells [124,125], hepatoma cells [126], and head and neck cancer cells [127]. Of note, treatment sequence determines whether YC-1 enhances or inhibits the effect of radiation [175]. Radiation followed by YC-1 leads to radioresistance due to YC-1-mediated increase in tumor hypoxia, while in the reverse order, YC-1 suppressed the postirradiation upregulation of HIF-1 activity and consequently delayed tumor growth.

PX-478 initially got attention due to its antitumor activity, such as suppression of cell growth and proliferation as well as induction of apoptosis, but back then the mechanism was elusive [128]. Later, it reveals that PX-478 is an effective HIF-1 inhibitor via a decrease of HIF-1α transcription and translation and an increase of HIF-1α degradation [176]. In the context of radiation, PX-478 enhances radiosensitivity of prostate carcinoma and hepatoma cells under hypoxic conditions by inhibiting HIF-1α expression [177,178]. In addition, PX-478 radiosensitizes glioma and pancreatic tumor through inhibition of HIF-1–dependent proangiogenic signaling [179,180]. In a phase I trial with 40 advanced-stage cancer patients [128] (NCT00522652), only a limited number of them experience severe events, and a relatively high proportion of patients (39%) achieve stable disease.

### 6.4. Hypoxic Radiosensitization by Inhibition of Tumor Metabolism

Altered energy metabolism is one of the hallmarks of cancer in which metabolism is shifted from oxidative metabolism towards glycolysis. This metabolic phenotype not only provides the building blocks to sustain unlimited proliferation of tumor cells but also generates abundant antioxidant to keep the redox balance [27]. Dichloroacetate (DCA), a synthetic small molecule used to treat hereditary metabolic or cardiovascular diseases, is an inhibitor of mitochondrial pyruvate dehydrogenase kinases. DCA, therefore, can modify tumor metabolism by activating mitochondrial activity to force glycolytic tumor cells into oxidative phosphorylation [181]. Treatment of medulloblastoma cells with DCA increases radiosensitivity that may link to the inhibition of glycolysis, the increase of ROS production, and the decrease of cancer stem cell-like characters [130]. Furthermore, DCA combined with radiation improves the survival of orthotopic glioblastoma-bearing mice, with mechanisms associated with cell-cycle arrest, increasing the oxidative stress as well as DNA damage [131]. Similar radiosensitization has been observed in lung [132] and prostate cancer cells [133]. Next, DCA increases the antitumor effectiveness of hypoxic cytotoxin such as tirapazamine without causing depression of hematologic parameters [182]. Currently, DCA is being tested in early clinical trials in patients with lung, head and neck, and brain cancers. In addition to DCA, suppression of glycolysis via ritonavir (glucose transporter inhibitor), 2-deoxyglucose (hexokinase inhibitor), and lonidamine (hexokinase inhibitor) are reported to enhance the response of the tumor to radiation, and they are under investigation in clinical trials in different types of cancer [134,135,136,183]. 

### 6.5. Hypoxic Radiosensitization via Reduction in Oxygen Demand

Oxygen is a natural radiosensitizer due to its effect of fixation of radiation induced DNA damage, and to form ROS, the effector molecules of radiation. Unsurprisingly, increase of oxygen delivery as a strategy to counter hypoxic radioresistance has been explored intensively, such as using hyperbaric oxygen. Alternatively, reduction in oxygen demand, that is, to decrease oxygen consumption, has drawn considerable attention recently, especially with clinically-relevant agents that are reported to overcome hypoxic radioresistance [184]. 

Glucocorticoids, a class of steroid hormones, increase tumor oxygenation via decrease of oxygen consumption, resulting in enhancement of tumor radiosensitivity by a factor of 1.7 [137]. Nonsteroidal anti-inflammatory drugs (NSAIDs) elevate tumor oxygenation via mediating mitochondrial respiration [138]. Subsequently, they improve radioresponse when radiation is applied at the time of maximal reoxygenation, which is comparable to the radiosensitization effect induced by hyperoxic gas breathing. Metformin, the most widely prescribed anti-diabetes drug, enhances response to radiation through improving tumor oxygenation via inhibition of mitochondrial complex I [139,140]; and the combination of radiation with metformin are under investigations in different clinical trials [141].

### 6.6. Others

Mitochondria are the primary site of ROS generation in cells, therefore, targeting enzymes located in mitochondrial electron transport chain holds a chance to perturb ROS homeostasis and overcome hypoxic radioresistance. Arsenic trioxide, a therapeutic agent against acute promyelocytic leukemia and certain solid tumors [185,186], is an effective inhibitor of mitochondrial complex IV [187]. In two murine models of radioresistant hypoxic cancer, arsenic trioxide decreases glutathione levels and increases intracellular ROS [142]. Subsequently, arsenic trioxide significantly reduces the hypoxic fraction of the tumor, resulting in a 2.2-fold increase in the response of tumors to radiotherapy [142]. In comparison with arsenic trioxide, an arsenic cytotoxin, darinaparsin, although demonstrates higher hypoxic radiosensitizing activities against solid tumor, the antitumor effects are associated with inhibition of oncogene rather than induction of ROS generation [188].

The introduction of nanotechnology, particularly heavy-metal nanomaterials with high atomic number (Z) values, provides new insight into the development of hypoxic radiosensitizers [143]. Among them, the most widely studied is gold nanomaterial due to its satisfying chemical stability, high biocompatibility, and low toxicity [144,145,146,147,189,190] and, importantly, its capacity to donate electrons to oxygen molecules to form ROS [148]. Under hypoxia, gold nanoparticles enhance colorectal tumor response to radiation of which is diminished by ROS scavenger [149]. In addition, integration of gold nanosphere and HIF-1α siRNA overcomes radioresistance of hypoxic tumors through excessive ROS generation and inhibition of DNA self-repair [150].

## 7. Conclusions and Perspectives

Hypoxia and ROS are two factors with opposite effects on the radioresponse of tumor: hypoxia is considered to be the most important cause of clinical radioresistance, while ROS are recognized as the primary cause of radiation-induced cell death. It was commonly considered that there is less oxidative stress in hypoxic tumor cells than normoxic counterpart owing to the shortage of ROS substrate oxygen. Thus, upregulation of ROS in cancer cells as a radiosensitizing strategy is always overlooked in the context of hypoxia. In fact, evidence reveals that hypoxic tumor cells generate more ROS via several mechanisms, including increased lifetime of ubisemiquinone that creates more opportunity for oxygen to react with the electron, and upregulation of iNOS expression. In line, NO generated by NO donors or activation of NOS, ROS generated by inhibition of antioxidant enzymes and glycolysis, and perturbation of ROS homeostasis by inhibition of HIF-1 enhance the radioresponse of hypoxic tumor cells, and some of the reagents are being tested in clinical trials.

With the growing knowledge of ROS pathways in hypoxic tumor cells, new light is shed on druggable targets in counteracting hypoxic radioresistance. For example, autophagy, a lysosomal degradation pathway, is activated in hypoxia and attributable to radioresistance due to clearance of hypoxia-induced ROS [43,191,192]. Inhibition of autophagy is reported to increase oxidative stress and cause the death of hypoxic tumor cells [26,193], while whether it could enhance hypoxic radioresponse is still largely unknown and deserves further investigation. Phosphoglycerate dehydrogenase (PHGDH) is the rate-limiting enzyme in the serine synthesis pathway, providing essential precursors for antioxidants synthesis. PHGDH is overexpressed in breast, melanoma, and cervical cancer patients and associated with poor outcome. Loss of PHGDH expression in hypoxic tumor cells disturbs mitochondrial redox homeostasis, resulting in increased apoptosis and abrogated breast cancer stem cells enrichment [194], making PHGDH an attractive target for hypoxic radiosensitization. Taken together, in the battle against hypoxic radioresistance, with emerging new insight in the interaction among radiation, hypoxia, and ROS, disruption of ROS homeostasis as a hypoxic radiosensitizing approach might hold the power to win the battle and deserves more attention.

## Figures and Tables

**Figure 1 cancers-11-00112-f001:**
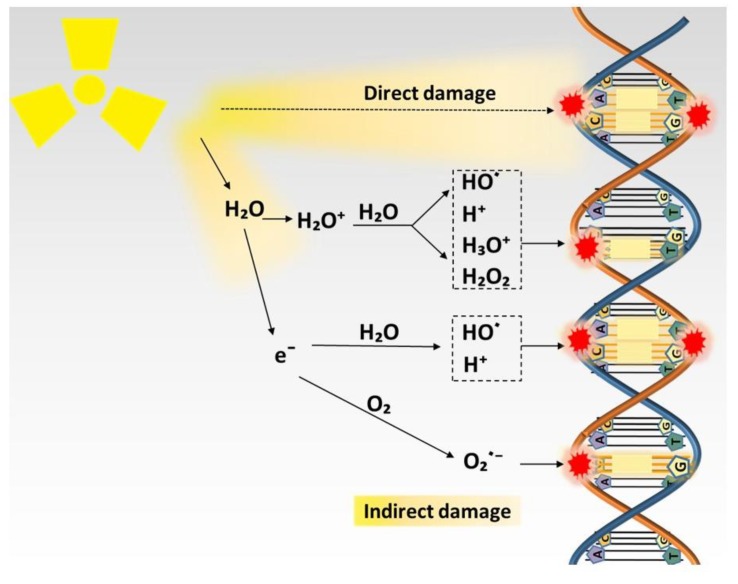
Direct and indirect actions of radiation. In direct reaction, radiation directly interacts with DNA resulting in DNA damage. In indirect reaction, radiation interacts with other molecules in the cells, particularly water, to produce free radicals such as hydrogen atoms (H^+^), hydroxyl radicals (HO), and superoxide radical anion (O_2_^−^), which in turn induce the damage to the DNA.

**Figure 2 cancers-11-00112-f002:**
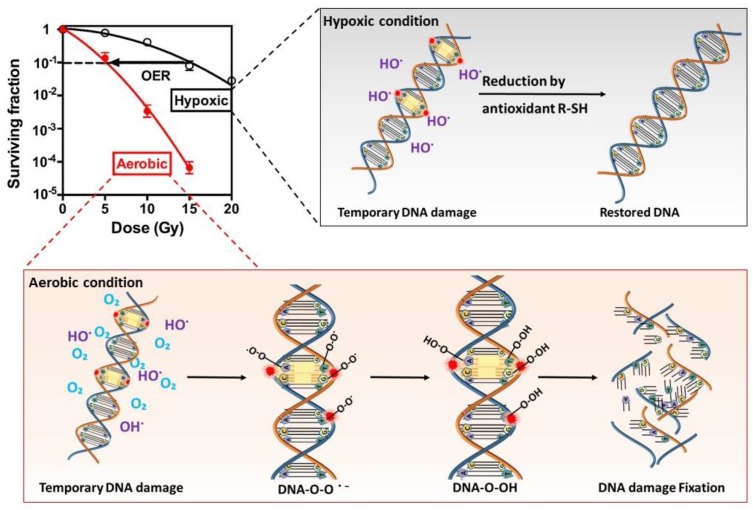
The oxygen fixation hypothesis. Under aerobic condition, radiation induced DNA radicals are able to react with oxygen, resulting in permanent DNA damage and strand breaks. Under hypoxic condition, the lack of oxygen enables the DNA radicals to be reduced to the original form that hampers the generation of strand breaks. Hypoxia-induced radioresistance can be estimated by survival curves. Briefly, the oxygen enhancement ratio (OER) or hypoxic radioresistance can be represented by a ratio, which is calculated by dividing doses administered under hypoxic to aerobic conditions needed to achieve a same survival fractions.

**Figure 3 cancers-11-00112-f003:**
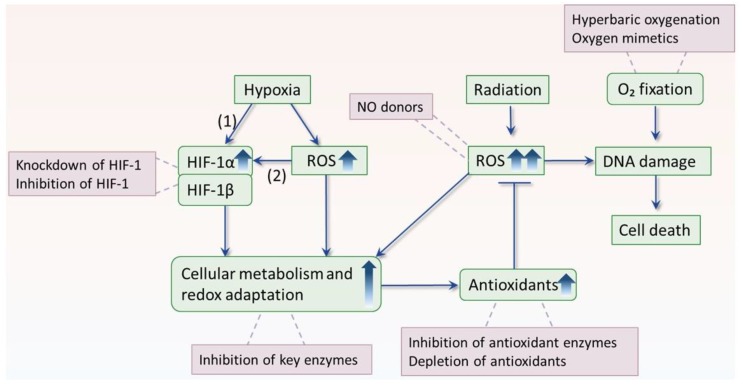
The interplay among hypoxia, ROS and radiation, and strategies to overcome hypoxic radioresistance. Radiotherapy kills cancer cells by causing DNA damage via generation of reactive oxygen species (ROS). However, under hypoxic condition, hypoxia induces HIF-1α accumulation by (1) prevention of protein degradation, or (2) upregulation of gene expression via ROS mediated pathways. As a result of increased HIF-1α, HIF-1 is activated and regulates more than a hundred of genes, conferring radioresistance by acting upon multiple mechanisms at different levels. For example, HIF-1 enhances expression of genes implicated in antioxidant defense systems, resulting in increased capacity to buffer ROS and radioresistance. In addition, hypoxia and radiation induced ROS could trigger a feedback loop that is in favor of generation of antioxidant. To counteract hypoxic radioresistance, historically, hyperbaric oxygen, and oxygen mimetic radiosensitizers have been explored, but failed in implementing in clinical practice due to their inconvenient application or side effects. Given ROS are the primary effector molecules of radiation, and hypoxic tumor cells strongly dependent on antioxidant defense systems to sustain ROS homeostasis, exposure of ROS insults to hypoxic tumor cells or perturbation of ROS adaptation pathway may lead to selective cytotoxicity and radiosensitization. In respect of this, approaches such as inhibition of HIF-1, suppression of antioxidant enzymes, and NO donors are under active investigation. The radiosensitizing approaches are indicated in red frames.

**Table 1 cancers-11-00112-t001:** Prognostic significance of hypoxia for irradiated cancer in different types.

Publication	No. of Patients	Oxygenation Parameter	Endpoint	*p*-Value *
**Cervix Cancer**				
Hockel et al., 1996 [45]	103	median pO_2_ < 10 mm Hg	DFS	=0.009
			OS	=0.004
Knocke et al., 1999 [46]	51	median pO_2_ ≤ 10 mm Hg	DFS	<0.02
Sundfor et al., 2000 [47]	40	subvolume pO_2_ < 5 mm Hg	DFS	=0.0001
			OS	=0.0004
			LC	=0.0006
Fyles et al., 2002 [48]	106	fraction pO_2_ < 5 mm Hg	PFS	<0.004
Nordsmark et al., 2006 [49]	120	median pO_2_ < 4 mm Hg	LC; OS	n.s.
**Head and Neck Tumors**				
Gatenby et al., 1988 [50]	31	pO_2_ < 5 mm Hg	LC	<0.001
Brizel et al., 1999 [51]	63	median pO_2_ < 10 mm Hg	DFS	=0.005
			OS	=0.02
			LC	=0.01
Stadler et al., 1999 [52]	59	subvolume pO_2_ < 5 mm Hg	OS	<0.01
Rudat et al., 2001 [53]	134	fraction pO_2_ < 2.5 mm Hg	OS	=0.004
Nordsmark et al., 2005 [54]	397	fraction pO_2_ ≤ 2.5 mm Hg	OS	=0.006
**Soft Tissue Sarcomas**				
Brizel et al., 1996 [55]	22	median pO_2_ ≤ 10 mm Hg	DF	=0.01
Nordsmark et al., 2001 [56]	31	median pO_2_ ≤ 19 mm Hg	OS	=0.01

(* multivariate analysis). (DFS = disease-free survival, OS = overall survival, LC = local control, PFS = progression-free survival, n.s. = not significant).

**Table 2 cancers-11-00112-t002:** Summary of hypoxic radiosensitizing reagents.

Name of the Agents	Mechanisms of Action	Cancer Types	References
**Hypoxic Radiosensitization by NO**			
Diethylamine nonoate	NO donor	Chinese hamster V79 lung fibroblast	[107,108]
S-Nitrosoglutathione	NO donor	Chinese hamster V79 lung fibroblast	[107]
Nitroglycerin	NO donor	Rectal cancer	[109]
Spermine nonoate	NO donor	Murine mammary carcinoma SCK	[110]
Sodium nitroprusside	NO donor	Human pancreatic tumor cells	[111]
Insulin	Activate eNOS	Liver and fibrosarcoma mouse tumors	[112]
Endogenous NO	Activate iNOS	Murine mammary carcinoma EMT6	[113]
**Hypoxic Radiosensitization by Inhibition of Antioxidant Enzymes**			
Buthionine sulphoximine + Misonidazole	Deplete glutathione and mimic oxygen	Multiple types of cancer cells	[114]
Buthionine sulphoximine + SR2508	Deplete glutathione and mimic oxygen	Multiple types of cancer cells	[114,115]
Dimethylfumarate	Deplete glutathione	Chinese hamster ovary cells	[16]
Diethylmaleate	Deplete glutathione	murine mammary carcinoma EMT6	[17]
DMF + Misonidazole	Deplete glutathione and mimic oxygen	Ehrlich ascites tumors	[116]
DEM + Misonidazole	Deplete glutathione and mimic oxygen	Multiple types of cancer cells	[114,116]
Piperlongumine	Inhibit glutathione S-transferase and thioredoxin reductase	Lung cancer cells	[18,117]
Auranofin	Inhibit thioredoxin reductase	Breast cancer cells and tumor models	[19,20]
Auranofin + BSO	Inhibit thioredoxin reductase and deplete glutathione	Breast cancer cells and tumor models	[19,20]
**Hypoxic Radiosensitization by Inhibition of HIF-1**			
HIF-1 siRNA	Silence HIF-1α	Hepatoma cells SMMC-7721 and prostate cancer cells PC3	[118,119]
SN-38	Inhibit radiation-induced HIF-1α	Colorectal cancer cells HT29 and SW480	[120]
Atorvastatin	Inhibit hypoxia-induced HIF-1α	Prostate cancer cells PC3	[121]
NSC74859	Inhibit HIF-1α and VEGF expression	Esophageal squamous carcinoma cells ECA109 and TE13	[122]
Berberine	Inhibit HIF-1α and VEGF expression	Prostate tumor models	[123]
YC-1	Inhibit HIF-1α translation and degrade HIF-1α	Multiple types of cancer cells	[124,125,126,127]
PX-478	Decrease HIF-1α transcription and translation and degrade HIF-1α	Multiple types of cancer	[128,129]
**Hypoxic Radiosensitization by Inhibition of Tumor Metabolism**			
Dichloroacetate	Inhibit glycolysis	Multiple types of cancer cells	[130,131,132,133]
Ritonavir	Inhibit glucose transporter	Head and neck carcinoma model HEP-2	[134]
2-deoxyglucose	Inhibit hexokinase	Glioblastoma	[135]
lonidamine	Inhibit hexokinase	Cervical cancer HeLa cells	[136]
**Hypoxic Radiosensitization via Reduction in Oxygen Demand**			
Glucocorticoids	Decrease oxygen consumption	Liver and fibrosarcoma mouse tumors	[137]
NSAIDs	Mediate mitochondrial respiration	Liver and fibrosarcoma mouse tumors	[138]
Metformin	Inhibit mitochondrial complex I	Multiple types of cancer	[139,140,141]
**Others**			
Arsenic trioxide	Inhibit mitochondrial complex IV	Liver and Lewis lung carcinoma models	[142]
Gold nanoparticles	Donate electrons to form ROS	Multiple types of tumor models	[143,144,145,146,147,148,149,150]

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
