# Peer review of "Hypoxic Radioresistance: Can ROS Be the Key to Overcome It?"

_cancers, 2019, doi:10.3390/cancers11010112_

Round 1
Reviewer 1 Report
In this manuscript by Wang and co-workers, the authors present a theoretical survey of the biological role of ROS in hypoxic radioresistance. Presenting the link between tumor hypoxia, ROS, and tumor responsiveness to radiation may be a challenging act owing to the multitude of biological pathways that indirectly or directly influence these parameters. Nevertheless, the presentation is lucid and generally well written although there are certain aspects that need clarification to improve the manuscript.
The Introduction is concisely written and puts the presentation into context. A flaw is that the authors evidently quote directly from a number of given references in a manner that results in leaping transitions from one sentence (with references) to the next (with its references). One example is the incomprehensible statement on lines 50-51: [quote] “Correspondingly, the antioxidant defense network is more mobilized to hypoxia, making hypoxic tumor cells more vulnerable to any ROS insults that impair links involved in sustaining ROS balance [23-24].” Whatever does this mean? This verbiage does not lead to the conclusion in the subsequent sentence, “Thus, the increase of oxidative stress…” and therefore needs thorough editing for scientific clarity.
There is a discrete number of typos throughout the manuscript that should be corrected.
The paragraph on biomarker investigations of tumor hypoxia (3.3) should be more inclusive. In particular, image-based modalities are limited to the description of PET. Over the recent years, innovative approaches within functional MR imaging have been published, and these should be summarized and discussed correspondingly.
Paragraph 5.1 starts with a statement (lines 211-212) that is not sufficiently underpinned by references: [quote] “…many studies have reported that chronic hypoxia elicits an increase in oxidant production… [21, 76].” Only two references are given, one review from 2006 and one study on AML cells. Since this is a central dogma of the review, it needs more substantial underpinning, preferably by original studies.
Moreover, paragraph 5.1 ends with the statement that [quote] “…acute hypoxia stimulates ROS production as well… [83]”, and the examples provided are from myocardial injury and hepatocyte pathophysiology (references 83-85). The authors are referred to a recent publication (PMID 30273860) that unambiguously demonstrated the combination of acute hypoxia and repressed ROS in colorectal cancer and how this may contribute to systemic inflammation and adverse outcome of radiotherapy in rectal cancer. The paper also discusses recent insights into the biological role of ROS and the mitochondrial metabolism in the host response to aggressive cancer. These viewpoints may be well worth implementing in paragraph 5.1 and taken into consideration of the review’s thematics.
Author Response
Response to Reviewer 1
Reviewer #1:
1. A flaw is that the authors evidently quote directly from a number of given references in a manner that results in leaping transitions from one sentence (with references) to the next (with its references). One example is the incomprehensible statement on lines 50-51: [quote] “Correspondingly, the antioxidant defense network is more mobilized to hypoxia, making hypoxic tumor cells more vulnerable to any ROS insults that impair links involved in sustaining ROS balance [23-24].” Whatever does this mean? This verbiage does not lead to the conclusion in the subsequent sentence, “Thus, the increase of oxidative stress…” and therefore needs thorough editing for scientific clarity.
For scientific clarity, the last several sentence of introduction have been rephrased.
Changes made can be found on lines 49-56.
2. There is a discrete number of typos throughout the manuscript that should be corrected.
The typos have been rectified.
Changes made can be found on lines 10, 34, 83, 90, 102, 122, 133, 204, 251, 310, 314, 397, 428.
3. The paragraph on biomarker investigations of tumor hypoxia (3.3) should be more inclusive. In particular, image-based modalities are limited to the description of PET. Over the recent years, innovative approaches within functional MR imaging have been published, and these should be summarized and discussed correspondingly.
MR techniques based approaches to measure tumor hypoxia have been briefly summarized in the section of 3.3.
Changes made can be found on lines 187-194.
4. Paragraph 5.1 starts with a statement (lines 211-212) that is not sufficiently underpinned by references: [quote] “…many studies have reported that chronic hypoxia elicits an increase in oxidant production… [21, 76].” Only two references are given, one review from 2006 and one study on AML cells. Since this is a central dogma of the review, it needs more substantial underpinning, preferably by original studies.
More articles regarding induction of ROS production by chronic hypoxia have been cited, and the findings from the original articles have been briefly discussed accordingly.
Changes made can be found on lines 225-241.
5. Moreover, paragraph 5.1 ends with the statement that [quote] “…acute hypoxia stimulates ROS production as well… [83]”, and the examples provided are from myocardial injury and hepatocyte pathophysiology (references 83-85). The authors are referred to a recent publication (PMID 30273860) that unambiguously demonstrated the combination of acute hypoxia and repressed ROS in colorectal cancer and how this may contribute to systemic inflammation and adverse outcome of radiotherapy in rectal cancer. The paper also discusses recent insights into the biological role of ROS and the mitochondrial metabolism in the host response to aggressive cancer. These viewpoints may be well worth implementing in paragraph 5.1 and taken into consideration of the review’s thematics.
Given that increased ROS production under hypoxia is the central part of the review, we have added evidence from several research articles to underpin the statement. The recent publication (PMID 30273860) recommended by the reviewer has been cited and briefly discussed as well.
Changes made can be found on lines 239-241, 257-259 and 264-274.
Reviewer 2 Report
The authors present a good review showing hypoxia induced ROS playing a crucial role involved in radioresistance. There are some suggestions to be addressed by authors before manuscript could be acceptable for publication.
Major issues
1. Studied with hypoxia and ROS has been mentioned in this manuscript, and the author also revealed the relationship between chronic hypoxia and ROS production. There were some studied pointed out that acute hypoxia had many effects than chronic hypoxia. This should be discussed in this review.
2. The authors enumerated some drugs or compounds which could inhibit antioxidant enzyme or HIF-1. It’s better to make a table to list these compounds for reading.
Minor issues
1. The resolution of figure 2 should be more clearly.
Author Response
Response to Reviewer 2
Reviewer #2:
Major issues
1. Studied with hypoxia and ROS has been mentioned in this manuscript, and the author also revealed the relationship between chronic hypoxia and ROS production. There were some studied pointed out that acute hypoxia had many effects than chronic hypoxia. This should be discussed in this review.
Indeed, acute hypoxia has much more pronounced effect to induce ROS production than chronic hypoxia. We overlooked this, the evidence from several articles have been added in and briefly discussed.
Changes made can be found on lines 257-259 and 264-274.
2. The authors enumerated some drugs or compounds which could inhibit antioxidant enzyme or HIF-1. It’s better to make a table to list these compounds for reading.
A table has been made to summarize all the compounds.
Changes made can be found in table 2.
Minor issues
1. The resolution of figure 2 should be more clearly.
The resolution of figure 2 has been improved, as well as the other two figures.
Changes made can be found on lines 79, 137, 305.
Reviewer 3 Report
In this manuscript, Wang and coworker summarized the hypoxic radioresistance and roles of reactive oxygen species (ROS). Firstly, the authors demonstrated biological action of radiation to induce DNA damage. Subsequently, mechanisms of hypoxic radioresistance and generation of ROS were shown. Then, the unexpected interplay between hypoxia and ROS, enhancement of hypoxia via ROS production and hypoxia adaptation mediated by ROS, was displayed. Finally, the authors discussed several methods overcoming hypoxic radioresistance. Because this review begins with basic principles of radiotherapy and hypoxia, it is very easy to understand. Thus, this reviewer recommends this manuscript to be published after a minor revision.
1. The authors introduced several methods to overcome hypoxic radioresistance. However, most of them focus on disruption of ROS homeostasis. Recently, there are several reports to overcome hypoxia via oxygen generation or oxygen delivery. Because they also have strong potential to enhance radiation therapy efficacy, it is recommended to be included in this review.
2. It would be very helpful to provide a table to summarize the radiosensitizers.
Author Response
Response to Reviewer 3
Reviewer #3:
1. The authors introduced several methods to overcome hypoxic radioresistance. However, most of them focus on disruption of ROS homeostasis. Recently, there are several reports to overcome hypoxia via oxygen generation or oxygen delivery. Because they also have strong potential to enhance radiation therapy efficacy, it is recommended to be included in this review.
Overcoming hypoxia via modulating oxygen levels in the tumor indeed is a very effective approach, to discuss about it, a new section has been added (6.5. Hypoxic radiosensitization via reduction in oxygen demand).
Changes made can be found on lines 435-450
2. It would be very helpful to provide a table to summarize the radiosensitizers.
A table has been made to summarize all the compounds.
Changes made can be found in table 2.
Round 2
Reviewer 1 Report
The authors have satisfactorily responded in the revised manuscript version to my criticism pertaining to the original submission.